# Molecular ensemble junctions with inter-molecular quantum interference

Ping'an Li[1] & Yoram Selzer ◉ [1] ✉

We report of a high yield method to form nanopore molecular ensembles junctions containing ~40,000 molecules, in which the semimetal bismuth (Bi) is a top contact. Conductance histograms of these junctions are double-peaked (bi-modal), a behavior that is typical for single molecule junctions but not expected for junctions with thousands of molecules. This unique observation is shown to result from a new form of quantum interference that is inter-molecular in nature, which occurs in these junctions since the very long coherence length of the electrons in Bi enables them to probe large ensembles of molecules while tunneling through the junctions. Under such conditions, each molecule within the ensembles becomes an interference path that modifies via its tunneling phase the electronic structure of the entire junction. This new form of quantum interference holds a great promise for robust novel conductance effects in practical molecular junctions.

An electron wave transmitted through a conductor's orbital acquires a phase, θ, which provides information complementary to the transmission probability $T = |t|^2$, with t being the transmission amplitude[1,2]: $t = \sqrt{T}e^{i\theta}$. Control and manipulation of this phase enables to modify transport properties by quantum interference (QI). In single-molecule junctions (SMJs), since the size of the metal contacts is comparable to $\lambda_F$, the Fermi wavelength within the leads[3–5], a single transverse mode ($N_{modes} = 1$) probes a single molecule ($N_{molecules} = 1$). This single incident electron wave at the Fermi energy, $E_F$, tunnels via the multiple molecular orbitals (MOs) of the molecule, each with a transmission probability of $T_k$, while experiencing a phase shift $\theta_k$, relative to the incident wave[6–12]. The overall transmission probability can then be described by[13]:

$$T(E_F) = \sum T_k(E_F) + 2\sum_{k>l} e^{i(\theta_k - \theta_l)} \sqrt{T_k(E_F)T_l(E_F)} \quad (1)$$

with the second term describing what we term here as intra-molecular QI that may either increase or decrease the probability of tunneling (Fig. 1a). If, for simplicity, the transmission probability depends on only two MOs, when there is no phase difference between these two paths then QI is constructive and $T(E_F)$ is high and if the two phases differ by $\pi$, then QI is destructive and $T(E_F)$ is low. Experimental verification of QI in SMJs[11,12] has been motivated by prospect harvesting of this

phenomena for applications such as highly efficient thermoelectric devices[14] and extremely small perfect insulators[15], as well as for inducing properties such as highly non-linear I-V characteristics[16]. The main challenge for the realization of these applications is precise control of the parity of the MOs that dominate the transport properties, as well as of their energy position relative to the Fermi energy[8].

In contrast to SMJs, in metal-based molecular ensembles junctions (MEJs), since $\lambda_F$ is also typically comparable to the distance between molecules[17–20] (Fig. 1b) the number of conductance modes is in the order of the number of assembled molecules ($N = N_{modes} = N_{molecules} \gg 1$). The overall conductance of the ensemble is then described by: $G(E_F) = \sum_{i=1}^{N} G_{eff}^i(E_F)$, where $G_{eff}^i$ is the effective conductance of each individual molecule. The individual conductance values can still be manipulated by intra-molecular QI, i.e., by varying their internal structure, however, since the molecules in MEJs are within close-packed ensembles, their $G_{eff}^i$ could also be affected by processes such as local crosstalk[21], electrostatic interactions[22,23] and dephasing effects[24].

While SMJs are a wonderful spectroscopic tool to probe the fundamentals of quantum transport in molecular junctions, it is clear that due to their limited stability, they are most likely not suitable for prospect applications and the future of molecular devices is more likely to be based on MEJs[17–20]. Motivated by this practical reasoning

[1]Department of Chemical Physics, School of chemistry, Tel Aviv University, Tel Aviv 69978, Israel. ✉e-mail: selzer@tau.ac.il

and since QI holds a great promise for many applications, we are looking for QI effects that go beyond the intra-molecular scale and instead utilize interference phenomenon within MEJs that emerge from the collective properties of the ensembles within these junctions. For this purpose, here we show that one approach towards this goal is to devise MEJs based on leads with a long $\lambda_F$ and consequently $N_{modes} \ll N_{molecules}$ (Fig. 1c). Under such junctions, each of the tunneling electrons through such junctions simultaneously and coherently probe ensembles of molecules in a process that can be termed inter-molecular QI. In this process, each of the interfering tunneling paths instead of being an MO, as in SMJs, is instead a molecule within the ensemble that modifies via its total tunneling phase the electronic

structure of the entire junction. This new tool to engineer QI conductance effects does not necessarily require molecules with pre-defined intra-molecular QI. We therefore demonstrate the existence of inter-molecular QI using a well-studied system of a self-assembled monolayers (SAMs) of alkane dithiols[20,25,26] by revealing a previously unresolved bi-modal behavior in their conductance properties. We argue that this bi-modal conductance behavior, which is typical to SMJs and entirely not expected in junctions with thousands of molecules, is a direct result of the sensitivity of the transport properties in these newly introduced MEJs to QI associated with the phases acquired by electron waves tunneling coherently and simultaneously through ensembles of molecules.

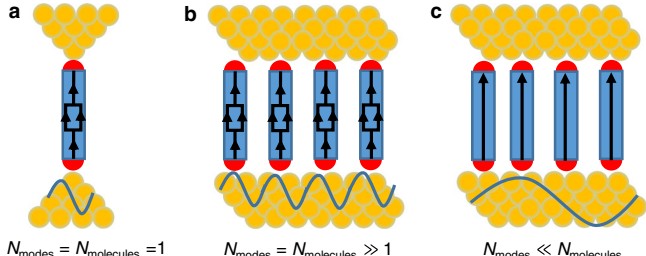

**Fig. 1 | Intra versus inter quantum interference. a** Intra-molecular QI is a result of the various interfering paths (MOs) of a single transverse conductance mode within the leads through a single molecule. **b** In MEJs, where the distance between molecules is comparable to the phase coherence length of the electrons at the Fermi energy, intra-molecular QI can prevail in each molecule but the ensemble as a whole does not contribute to the interference properties. **c** Inter-molecular QI takes place in MEJs based on leads in which the electrons has a sufficient long $\lambda_F$ to coherently probe by tunneling large ensembles of molecules. Interference in this case depends on the tunneling phase of each of the molecules within the ensembles.

## Results

### Fabrication and characterization of junctions

The junctions are based on a nanopore structure[27] with Au as a bottom contact and the semimetal Bi as a top contact. The two key properties of Bi that are essential for this study are its small density of states $DOS_{Bi} \sim 4.2 \times 10^{-6}$ states $eV^{-1}$ atom$^{-1}$ at the Fermi level[28] and its long Fermi wavelength of $\lambda_F^{Bi} \sim 30nm$[29]. Figure 2a depicts a schematic pre-sentation of a typical junction, and Fig. 2b–d describe concisely their fabrication protocol. The SAMs in all junctions are based on alkane dithiols, $HS\text{-}(CH_2)_n\text{-}SH$, with n=6, 8, 10 assembled on a Au bottom lead (see Methods). It is very unlikely that crosstalk exists between such molecules once assembled, which is important since such a process could potentially alter or totally mask inter-molecular QI[21].

Considering the size of the junctions (see typical AFM image in Fig. 2c) and using a density of ~4 molecules nm$^{-2}$ in SAMs on Au[20], the number of molecules in each junction is estimated to be ~40,000.

The are two reasons for the use Au as the bottom lead in all junctions: (a) The density of alkane-thiol SAMs on this metal is known[20] and as will be shown below is important for the quantitative analysis of

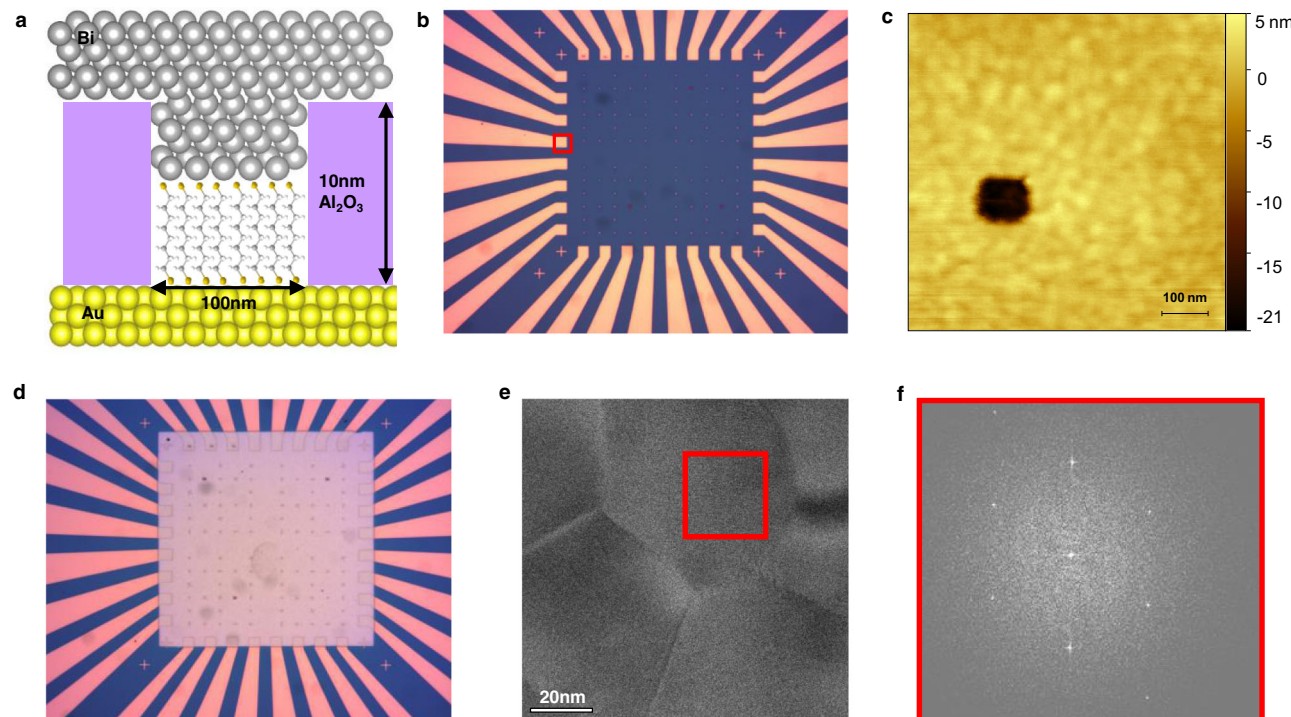

**Fig. 2 | Structure and fabrication of the MEJs.** The fabrication scheme is outlined in Methods. **a** Schematic presentation of a junction with its (not drawn to scale) dimensions. **b** An optical image of a device containing 31 junctions. The Au features are covered by an $Al_2O_3$ layer and a pore is formed at the end of each Au lead (within an area such as the one marked by the red rectangle). **c** An AFM image of one pore with its characteristic dimensions. **d** A device after Bi evaporation via a shadow mask. **e** A TEM image of a Bi layer deposited under the same conditions used for the top contact to the junctions. The grains of the layer are all several tens of nan-ometers in size. **f** The diffraction pattern of the red rectangle in **e**, which is typical for all measured grains, reveals Bi metal with a <111> orientation.

the bi-modal behavior observed in the junctions. (b) The coupling of the molecules to the leads, which in turn affects their conductance, depends on the density of states in the leads. The higher density of states in Au as compared to Bi, results in measurable conductance values. The use of Bi for both leads is expected to result in too-low conductance values when alkane chains are used.

The use of Au with its short Fermi wavelength of ~0.5 nm, on one side of the junctions, does not affect the inter-molecular QI in the junctions. The contact resistance between two leads with $N$ and $M$ transverse modes is: $R \sim \frac{h}{2e^2}\left(\frac{1}{M} + \frac{1}{N}\right)$, where $h$ is Planck's constant and $e$ the charge of an electron[30]. Since for the given geometry of the junctions the number of modes in Bi is orders of magnitude smaller than in Au (see discussion below), the entire behavior of the junctions is governed only by the Bi lead. Simulations (see Methods) also verify this argument.

As the effective length of $\lambda_F^{Bi}$ is critical to the new approach of inter-molecular QI, the crystalline structure of the top Bi contact needs to be of high quality, i.e., without any defects or contaminations which could decrease $\lambda_F^{Bi}$. This has been verified by analysis of transmission electron microscope (TEM) images of Bi films evaporated under the same conditions used for the junctions, which show large grains (Fig. 2e), all with a < 111> structure devoid of any oxygen contamination (Fig. 2f).

The yield and electrical quality of the junctions were initially evaluated by electrical measurements at room temperature under vacuum, following a previously suggested sorting protocol[31]. Junctions with a conductance larger than the quantum of conductance, $G_0 = 2e^2/h$, were discarded on the basis that their conductance is most likely due to a direct Bi-Au contact. This reasoning for the initial sorting process holds only for measurements performed at room temperature since at cryogenic temperatures atomic-sized Bi contacts could have a sub-$1G_Q$ conductance due to quantum confinement[32]. All junctions chosen by the above initial criteria, were further analyzed by characterizing the linearity of their I-V curves[31], using their resistance at two bias values, $R_{50mV}$ and $R_{300mV}$. Curves with $\delta=|1-R_{50mV}/R_{300mV}|<0.1$ were considered linear and their respective junctions were not treated further. The yield of molecular junctions based on this initial sorting process was found to be: 63% (C6), 53% (C8) and 57% (C10).

Inelastic electron tunneling spectroscopy (IETS)[31,33] was used to further assess the molecular nature of the junctions. Figure 3a shows the characteristic C-H stretch of alkane chains (360mev), which appears in all measured junctions. The fingerprint region of the junctions, shown in Fig. 3b for representative C8 and C10 junctions, appears to be quite reproducible although with some variation in the intensity of the peaks. The assignment of the vibrational modes in this region is based on previous studies[31,33]. The lack of any temperature effect on the conductance of junctions in the range between 8–300 K is shown in Fig. 3b. This behavior is similar to previous temperature-dependent measurements of alkane-chains[34] and is expected for the non-resonant coherent tunneling process in such a system. It also proves that no direct Bi-Au contacts were formed in the junctions, since for such contacts due to quantum confinement within the Bi[29,32], the ~$4k_B T$ smearing around the Fermi energy, should affect (increase) the conductance in this temperature range.

**A bi-modal conductance behavior of the junctions**

Figure 4a plots histograms of the low bias conductance values for all accepted junctions. The large volume of statistics presented in this figure is essential for the analysis described below and is a direct result of the high yield and reproducibility of the fabrication approach. The histograms show a double-peak distribution (a bi-modal behavior), with high, $G^H$, and low, $G^L$, conductance average values, which become better resolved as the length of the molecules increases, i.e., their statistics is a molecular property. Integration of the peaks reveals that

the percentage of the $G^H$ junctions in the entire population is 6%, 36% and 48% for the C6, C8 and C10 junctions, respectively. This rules out a possible explanation that the $G^H$ junctions are due to direct Bi-Au contacts as then, they should have become statistically less meaningful as the length of the molecules increases, i.e., as the SAMs become better ordered[20] and penetration of the evaporated Bi becomes less probable.

The fact that the conductance statistics of junctions with 40,000 molecules is bi-modal is intriguing and has never been reported before. Multi conductance values usually characterize SMJs[35–38] or junctions with highly ordered and extremely small ensembles of molecules (with less than 100 molecules) arranged on single crystal structures[39]. In such junctions, the multi-peak histograms are rationalized by variations in, for example, the coordination of the head group or in the case of alkane chains in SMJs, by the number of gauche defects formed in each measurement event[35]. In contrast, when the number of participating molecules is large, as in nanopore junctions containing alkane chains with a similar size as used here[31], the reported conductance histograms have only one average value for each molecule and the information on structural variations and their effect on conductance is hidden within the width and skewness of the histograms[40] but never as several peaks.

An explanation for the bi-modal behavior resulting from a change in the transport mechanism, i.e., that $G^H$ is due to triggering of a thermally activated process such as hopping, can be excluded by considering the lack of temperature effect shown in Fig. 3b. In addition, the similar off-resonance tunneling attenuation factor of $\beta$ ~1 $n^{-1}$ extracted for both sets of conductance values and shown in Fig. 4b is similar to previously reported values for of alkane-dithiols[41,42] and implies coherent off-resonance conductance for both $G^H$ and $G^L$.

## Discussion

We claim that the bi-modal behavior is a result of and proof for the existence of inter-molecular QI in the measured junctions. The proof starts by comparing between the values of $G^H$ and $G^L$ for each molecule and expected conductance values based on previous SMJs measurements[35–38,43–47], in which conductance histograms also unambiguously confirm sets of conductance values of $G^H_{SMJ}$ and $G^L_{SMJ}$, resulting most likely from combinations of different sulfur-metal couplings and trans/gauche conformations. Making a reasonable assumption that crosstalk between alkane chains is negligible[21], the conductance of the MEJs should simply be: $G^{H/L}_{MEJ} = \sum_{i=1}^{N} G^{H/L}_{SMJ}$ where $N = 40,000$. Justification for the accuracy of the estimated number of molecules in the junctions can be found in Methods.

In all previous SMJs measurements, the leads were made of Au. In order to account for the fact that here Bi is one of the contacts, we use the following expression for the low bias conductance of an SMJ[48]: $G_{SMJ} \cong G_0 \frac{\Gamma_L \Gamma_R}{\varepsilon_0^2}$, where $\varepsilon_0$ is the energy (relative to the Fermi level) of the 'gateway' molecular level[35,47–51] and $\Gamma_L$, $\Gamma_R$ are the coupling of this level to the left and right leads, respectively. Since these coupling terms are proportional to the DOS in the leads[52], a change of one lead from Au with DOS$_{Au}$~0.1 states eV$^{-1}$ atom$^{-1}$ to Bi should decrease the previously reported values of $G^{H/L}_{SMJ}$ by the ratio DOS$_{Bi}$/DOS$_{Au}$, i.e., by a factor of ~24,000. Based on this calculation, Fig. 4c shows a very good agreement between the experimental $G^L$ and the estimated $G^H_{MEJ}$ values for all molecules. This similarity implies that the measured two orders of magnitude difference between $G^H$ and $G^L$ for each molecule, is not due to the difference between their corresponding $G^H_{SMJ}$ and $G^L_{SMJ}$ values as observed in previous measurements[35–38,43–47]. It also implies that the average conformation and bonding geometry to the contacts for the alkanes in the SAMs is similar to those which are responsible for $G^H_{SMJ}$. Indeed dithiol-alkanes are reported to be better organized and with a higher fraction of all-trans molecules than alkane-thiols, most likely due to interactions between the exposed –SH groups[25,26,34]. This also indicates that the difference between the $G^L$ and $G^H$ junctions is not due

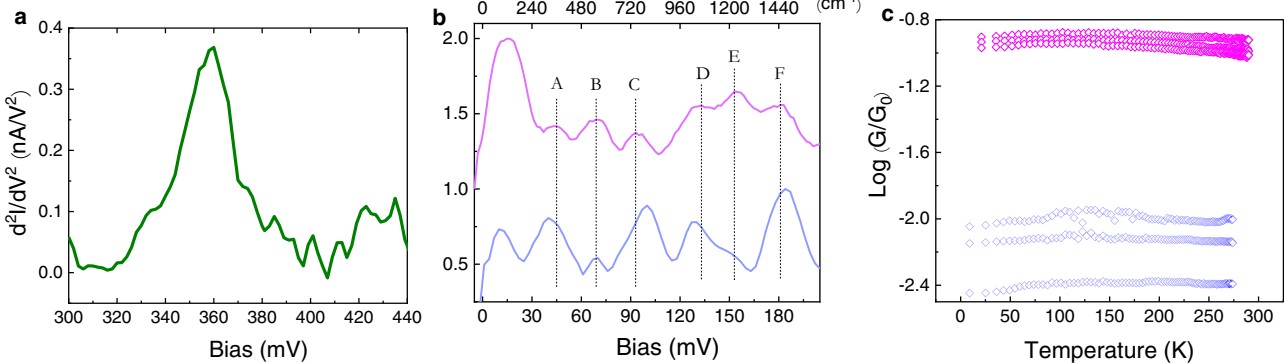

**Fig. 3 | Characterization of the MEJs. a** All IETS spectra show the characteristic C-H stretch of alkane chains at 360 meV. **b** IETS in the fingerprint region of representative C8 (magenta) and C10 (blue) junctions, with the following assignment of modes[31,33]: (A) Bi-S vibration, (B) C-S wag, (C) C-S stretch, (D) C-C stretch, (E) CH$_2$ wag, (F) CH$_2$ scissor. **c** The effect of temperature on the conductance of three different C8 (magenta) and three different C10 (blue) junctions in the temperature range between 8–300 K. The variation between junctions comply with the variation in conductance as presented in all other figures. The presented behavior for each junction was found to be highly reproducible under repeated measurements and temperature cycles.

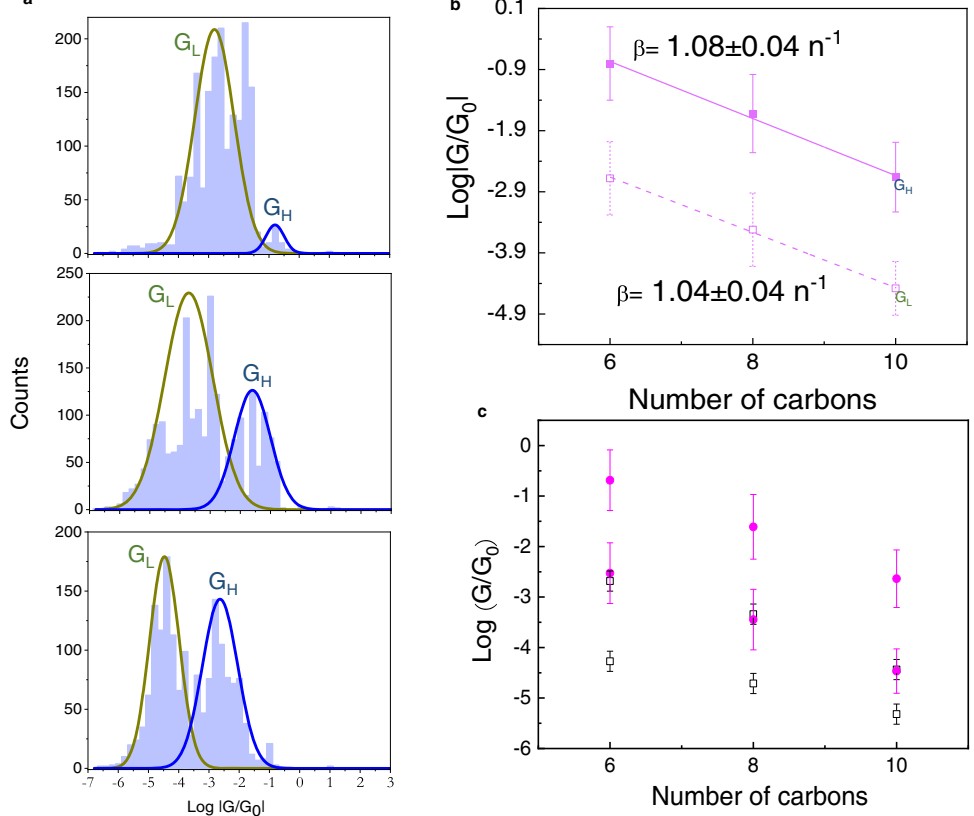

**Fig. 4 | Conductance histograms, tunneling attenuation factor and comparison to SMJs with the same molecules. a** Low bias conductance histograms measured for all accepted junctions. As the length of the chains increases, the high ($G^H$) and low ($G^L$) conductance peaks become better resolved. **b** A semi-logarithmic plot of $G^H$ and $G^L$ versus the number of carbons in the molecules, $n$. **c** Comparison between the normalized (see text for details) experimental $G^H$ and $G^L$ values (magenta circles) to the expected conductance-values (black rectangles) based on measurements of SMJs. The error bars of the new results are the standard deviations of the Gaussians in Fig. 4b. The error bars of the SMJs results were taken from the published results appearing in the references mentioned in the text.

to a lack of bonding at the top (Bi) contact in the former junctions, since $G^L$ peaks correspond to $G^H_{SMJ}$ values which were measured in junctions with contacts on *both sides*. Therefore, based on the above reasoning we conclude that the difference between the $G^L$ and $G^H$ junctions is not due to differences in the conductance of the individual components of the SAMs, but instead has to be a collective effect within the ensembles, i.e., due to inter-molecular QI.

The results in Figs. 5, 6 are consistent with Inter-molecular QI. The calculations (see Methods) are based on a two-dimensional model

junction (Fig. 5) in which each two-site molecule is coupled to the leads by a coupling term $-t_0$ and also has a phase-sensitive intra-molecule coupling term $-t^{i\theta}$, which affects the overall phase of the transmission through the molecule. This minimal model captures the properties of the thiol-alkane SAMs used here, in which the phase of the transmission through each molecule could be different (see discussion below), and at the same time the conductance of each molecule is the same and is phase-independent. Figure 5 depicts two junctions, which differ in the number of molecules with a transmission phase of $\theta = \pi$, $n_\pi$, and $\theta = 0$,

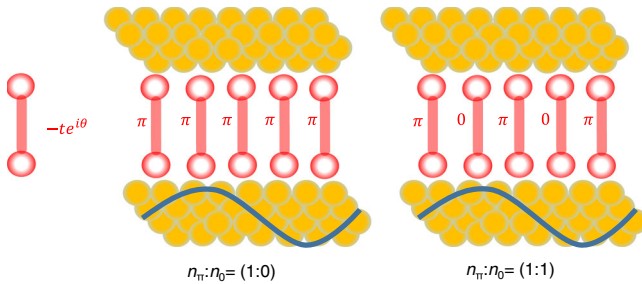

$n_\pi : n_0 = (1:0)$

$n_\pi : n_0 = (1:1)$

**Fig. 5 | The minimal 2D model used for simulations.** Schematic presentation of the minimal model used in the calculations based on two-site molecules assumed to be coupled to both leads with the same coupling strength and with a phase-dependent (intra) coupling term between sites. The phase, θ, can be either π or 0. The junctions used in the calculations contained 100 molecules. The image shows a junction with identical phase (π) for all molecules (left junction) and a junction with a 1:1 phase-ratio (right junction).

$n_0$. Figure 6a plots the effect of the ratio $n_\pi : n_0$ and the number of conductance modes at the Fermi energy on the conductance of the model junctions. Since in all presented junctions, all molecules have the same conductance regardless of the transmission phase, the apparent changes in the overall conductance along each curve are only due to the variation in the phase ratios. With only one conductance mode in the leads (bottom curves in Fig. 6a), destructive QI around $n_\pi / n_0$ ~1, results in junctions with a lower conductance as compared to the junctions with pure-phase ensembles of either $n_\pi$ ~1 or $n_0$ ~1. The overall conductance in these two last cases is identical. With increasing number of conductance modes, destructive QI in junctions with $n_\pi / n_0$ ~1 continues to prevail, however, with a diminishing effect (see Fig. 6b). As long as $N_{modes} \ll N_{molecules}$, in other words, as long as there are conducting electrons with sufficiently long wavelength to (coherently) tunnel through more than a single molecule, the transport phases of the participating molecules affect the overall conductance. When $N_{modes} = N_{molecules}$ (top curve in Fig. 6a), inter-molecular QI disappears (completely randomized) and conductance becomes phase-independent.

The above calculations suggest that for an ensemble of molecules, each with the same individual conductance value but with a possible different transmission phase, the overall conductance could vary as a function of the distribution of phases providing the ensemble is probed by electrons with a long coherent length that interfere with a sufficiently large fraction of the molecules within the ensemble.

The justification of this model to the junctions measured here, starts with the fact that their conductance without molecules (direct Bi-Au contact), was found to be $13G_0$. Under the conditions by which the film was made, Bi tends to form large and well-ordered <111> grains where only few of those are needed to cover the entire surface of the junctions (see Fig. 2e). With a characteristic mean free path in Bi that is much larger than the involved dimensions[29], the Sharvin[53] formula can be used to estimate the number of modes in the junctions, which for a junction with an edge length of $L$ is $M = 4L^2 / \pi \lambda_F^2$. With $\lambda_F^{Bi} = 30$ nm, and $L = 100$ nm, the number of modes should be $M = 14$, i.e., within good agreement with the measurements. Since the number of molecules in each junction is ~40,000, the ratio (in one dimension in order to allow comparison with the simulations) between this number and the (experimental) number of modes is $N_{molecules}/N_{modes} = \sqrt{40000/13} \sim 55$, which according to the calculations presented in Fig. 6a, is obviously sufficient for the contribution of QI not to be randomized. Since, one may regard this ratio as the effective number of molecules that is coherently and simultaneously probed by the tunneling electrons in these junctions, the experimental ratio of ~55 corresponds to the 2-modes (second from the bottom) curve in Fig. 6a, which was calculated for 100 molecules.

In contrast to conjugated molecules for which the phases associated with their pi-electrons scattering paths have been studied theoretically quite extensively[6–10], the mechanisms affecting the overall phases of electrons tunneling through alkane chains still need to be studied. Based on previous studies, which only implicitly discuss the role of phases, three possible mechanisms that can randomize the transmission phase (change the $n_\pi / n_0$ ratios) in alkane-chains within SAMs can be invoked. The first could be a result of inter-chain through-space tunneling[54–56]. As this process, by nature, is exponentially suppressed with distance, it is most likely taking place between chains via the closest carbons. Each such event adds an additional transport path longer by one carbon unit and consequently also a change of phase by $\pi$[57]. Based on molecular dynamics simulations[58,59], the contribution of this mechanism becomes higher for shorter chains due to a broader distribution of the tilt angles and with it a higher probability for inter-chain hopping[54]. A second randomizing mechanism can be associated with a conformational degree of freedom of the head groups in alkane chains[60,61]. The third is due to the conformation along the chains. While long (>C10) alkane chains are organized in ordered layers with ideally an all-trans conformation, shorter chains have a higher number of 'defects' in the form of cis or gauche conformations[62]. These conformations are argued to lower the conductance of chains because of destructive QI between through-bond and through-space tunneling paths in each molecule[63,64]. This QI, essentially between clockwise and anti-clockwise transport, should affect the overall phase of the tunneling electrons with a magnitude that depends on the di-hedral angle within the gauch conformation[65].

All the above phase-randomizing mechanisms most likely prevail within the short-chain SAMs used here, and as in any other case of a conducting system experiencing a transition from being one-dimensional to three-dimensional, it is expected to experience total randomization of the phases of all conducting paths[66], i.e., the system most likely resides around the $n_\pi : n_0$ ~1 ratio.

Returning to Eq. 1., the red curve in Fig. 7, shows the calculated conductance based on this equation of an MEJ with $N_{molecules} = 1000$ and one transverse mode at the Fermi level as a function of the $n_\pi : n_0$ ratio ($N_{molecules} = n_\pi + n_0$). In this calculation, since there is only one transverse mode, $N_{molecules}$ defines the number of interfering transmission paths. It is then easy to see by performing the summations in Eq. 1 up to $N_{molecules}$, that when the phase is identical for all paths (a situation which occurs on both ends of the red curve) the conductance is proportional to $N_{molecules}^2$. For this reason, for the purpose of calculation, the conductance of a single molecule was set to $10^{-6} G_O$, so that the total conductance for one mode does not exceed $1G_O$. The 'noise' around the dip in the red curve was inserted by varying the phase around π to be with a standard deviation of 10%. Thus for each given number of $n_\pi$ molecules, the phase of each molecule was set to be randomly distributed within the range of $\pi \pm 0.1\pi$. This simulates, as mentioned above, the dependency of phase on the dihedral angle between neighboring carbons[65]. The dashed green line is the conductance based only on the first term in Eq. 1 and it intersects the red curve at the point where QI is nullified, i.e. at $n_\pi : n_0 = 1$. The conductance at this point corresponds to the measured $G^L$ values (see Fig. 4c), which after normalization are identical to the conductance of SMJs with the same molecules, where (the overall) phase has no effect on the transport properties.

Any change in the phase ratio from this point, i.e., any increase in the structural order and uniformity within the SAM, leads to a substantial increase of the overall conductance due to constructive QI. The green and blue Gaussian curves show the requested phase-population change in order to result in a bi-modal conductance behavior with a two-orders of magnitude difference between the conductance of the two peaks, as observed experimentally. The possibility to shift the population between these two Gaussians depends on the length of the molecules. Thus, in MEJs with C6, that have a

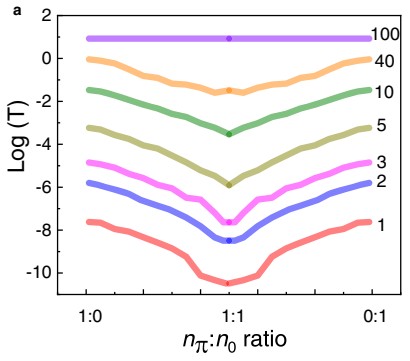
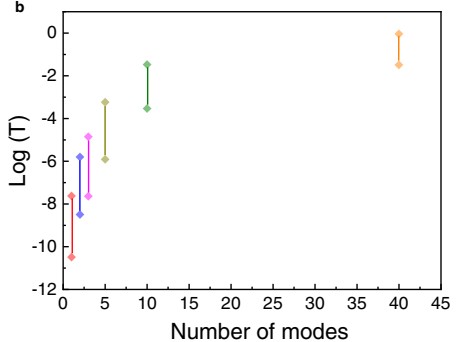

**Fig. 6 | The effect of phase randomization and the number of transverse modes on the transport properties of MEJs. a** Calculated transmission for junctions with the indicated number of conductance modes as a function of the phase ratio. The conductance of pure π and pure 0 phases, are identical. **b** The range of conductance variation as a function of the number of modes. With increasing number of modes, the difference between minimum conductance under complete phase randomization and maximum conductance under uniform phase, becomes smaller.

pronounce liquid-like structure[20] and therefore a phase that is only occasionally less randomized, the conductance is typically in the $G^L$ regime (where the red and green curves intersect) and most of the junctions are within the green population Gaussian. For this reason, it is experimentally very rare (see Fig. 4a) to find C6 junctions with a $G^H$ conductance. As the length of the molecules increases, there is statistically a higher probability to find better-ordered layers (blue Gaussian) with less phase randomization and a higher number of phase-identical molecules and as a result a larger population of junctions with conductance in the $G^H$ regime. The rough calculation based on Eq. 1 suggests that for a two orders of magnitude change in the conductance between $G^L$ and $G^H$, a change of ~15% is needed in the $n_\pi$:$n_0$ ratio. Molecular dynamics calculations suggest that the percentage of cis/gauche conformations at the end of long alkane chains at 300 K is in the same range, suggesting that indeed disorder in this scale is natural in alkane chains[58]. One should consider the red curve in Fig. 7 as the equivalent of the transmission curve (as a function of energy) of an SMJ with a destructive QI effect in its transport properties[6]. In the latter, if the Fermi energy resides at or very close to the transmission dip, any small change in the position of the Fermi energy results in a large change of conductance due to less destructive QI. Similarly, here, any structural shift of the monolayer within an MEJ from a full phase-randomized state leads to a large increase of conductance because of less destructive QI. This large change in conductance, resulting from a small change in (an average) structure, results in the bi-modal behavior observed here. MEJs with leads having a long Fermi wavelength, although containing thousands of molecules, essentially behave like junctions with very few molecules, in which each 'effective molecule' is a small ensemble probed by individual electrons having a long characteristic wavelength. One can consider this as a loss of resolution: with increasing wavelength of the tunneling electrons, the individual transmission properties of the molecules cannot be resolved anymore and the overall conductance starts to depend on the size of the ensembles that are being tunneled-through which in turn depends on the number of modes. When the number of "effective molecules" is sufficiently small the junction starts to behave similarly to a small-ensemble junction[39] with a bi-modal conductance behavior.

Figure 4a suggests that with increasing length and better ordered SAMs, one would expect a diminishing contribution of the $G^L$ peak, leading eventually to only one $G^H$ peak. In the case of alkane chains, such a behavior is expected for molecules longer than C10. However, due to the low density of states of Bi, the conductance of C10 junctions is already quite similar to the conductance of single molecule (with Au leads) and further increase of their length is expected to result in signals that are much more difficult to measure. Further studies on the effect of order necessitates junctions based on more conducting molecules and will be the subject of future work.

To conclude, the above results present two important steps towards the realization of practical molecular devices based on MEJs. Both steps rely on the unique properties of Bi. The first, in terms of fabrication, utilizes its low heat of evaporation to form top contacts onto ensembles of molecules with low damage statistics and what appears to be a high number of 'properly contacted' molecules. We expect that careful control of the roughness of the underlying substrates, use of smaller pores and molecules that are more rigid should make the yield of this process even higher. This in turn, could also enable the integration of molecular junctions into nano-scale circuits. The ability to form a large number of valid junctions has been utilized in this study to produce the necessary statistics to support the second advancement presented here which is the utilization of the long Fermi wavelength of Bi to enable a never before observed effect of inter-molecular QI in MEJs. This effect could be harnessed for the realization of highly stable and efficient thermoelectric, switches and non-linear molecular devices.

## Methods
### Fabrication and characterization of junctions
Junctions were formed by first depositing using an atomic layer deposition tool a 10–20 nm of an $Al_2O_3$ layer onto Au features, which served as bottom leads (Fig. 2b). After defining by e-beam lithography 100 nm × 100 nm squares on top of this layer, reactive ion etching was used to form the pores of the junctions. The structure of the resulting pores was analyzed by AFM (Fig. 2c) in order to ensure complete etching of the pores down to the underlying Au substrate. Monolayers within the etched pores were formed by a self-assembly process in which the samples were immersed overnight in 1 mM de-aerated solutions of the various molecules in ethanol under inert (glove-box) conditions. Bi was thermally evaporated via a shadow mask under a vacuum of <$10^{-6}$ torr, at a rate of 0.2 Å sec$^{-1}$. The thickness of the Bi films was 60 nm. Recent calculations show that the oxide layers, which naturally formed on the top of these films, do not affect their electronic properties[69]. In each device, such as the one shown in Fig. 2, the $Al_2O_3$ film was completely removed from one of the underlying Au features in order to establish direct contact to the top Bi layer. Probing of each junction was made through its bottom lead and its common Bi layer via the directly connected Au lead.

The apparent high yield of the above fabrication route can be attributed to two parameters: (i) The low heat of evaporation (105 kJ mol$^{-1}$) of Bi, which is lower than that of all previously used directly evaporated metals: Au (334 kJ mol$^{-1}$), Pb (178 kJ mol$^{-1}$), Al (293 kJ mol$^{-1}$), Pd (357 kJ mol$^{-1}$) and Ti (421 kJ mol$^{-1}$). It is therefore highly conceivable that the evaporated flux of hot Bi atoms onto SAMs is less deteriorating than in all previous attempts to use the same process with the above-mentioned metals. (ii) The negligible penetration of the evaporated

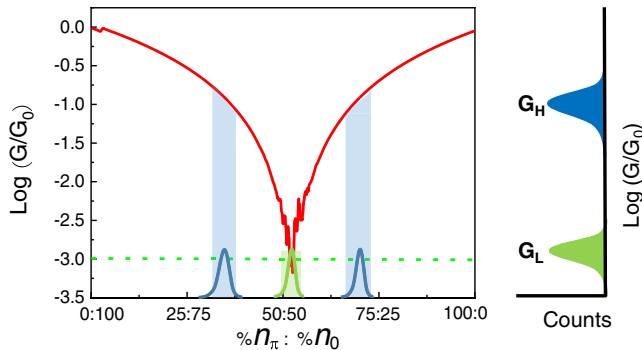

**Fig. 7 | The effect of phase randomization on transport.** Calculated conductance of a 2D MEJ with one transverse mode at the Fermi level based on Eq. 1. The red curve, calculated for $N_{molecules}$ = 1000 shows how the total conductance changes with the $n_\pi$:$n_0$ ratio. When the phase is uniform on both ends of the curve, the conductance (see Eq. 1) is proportional to $N_{molecules}^2$. The dashed green line is the conductance based only on the first term in Eq. 1. The green and blue Gaussian curves show the requested phase-population change in order to result in the bimodal conductance behavior shown on the right with two conductance peaks separated by two orders of magnitude.

atoms into the SAMs due to the use of dithiols, i.e., by using an end-group at the vacuum side, which has a high affinity towards Bi. Importantly, the junctions were not cooled down during evaporation, since cooling (under the conditions used here) causes condensation of water molecules on top of the monolayers, which makes the number of properly contacted molecules unknown. In addition, the cross section of the junctions was not made to be smaller than the average grain size of Bi, an approach usually taken in order to increase the probability for pin-hole free monolayers. The reason for this decision was that under such conditions accumulation of atoms on the walls of the pores usually block evaporated atoms from reaching the monolayers, resulting again with an unknown number of molecules within the junctions. As a result of all the above experimental conditions, the behavior shown in Fig. 4c is realized, i.e., that the normalization of the conductance of the junctions by the value of the approximated number of molecules works well for all molecules. This comes as another proof for the reproducibility of the fabrication scheme, suggesting that (almost) all assembled molecules are 'properly contacted'.

Inelastic electron tunneling spectroscopy measurements were performed at 8 K using a lock-in detection scheme with an *ac* modulation voltage of 10 mV at a frequency of 773 Hz.

Temperature-dependent conductance measurements were performed by a lock-in technique constantly probing the conductance of junctions around a dc bias of 0 V, with a modulation ac voltage of 3 mV at 137 Hz while slowly varying the temperature between 8–300 K over a time duration of ~20 hours.

### Simulations of the conductance of the 2D junctions
Detailed explanations how to calculate the conductance of ensembles of molecules based on the non-equilibrium Green function formalism are described in detail in ref. 24,67,68,. Here, 100 two-site molecules have been used with a phase-dependent coupling between the sites that is also phase dependent[14]. The number of modes was determined by the position of the Fermi energy within the dispersion of the leads[30,66]. For each number of modes, 100 simulations were performed with random numbers of $n_\pi$ (and consequently $n_0 = 100-n_\pi$), where the position of the molecules also randomly chosen, in order to consider the spatial form of the modes. The results shown in Fig. 5b, are the average of the conductance values which were calculated for each $n_0$: $n_\pi$ configuration, hence the fluctuations observed in the curves.

The electronic structure of the 2D molecular junction is represented by a tight-binding Hamiltonian $H=H_L+V_L+H_M+V_R+H_R$, where

$H_{L/R}$ and $H_M$ represent the left/right leads and the central molecular layer, respectively. $V_{L/R}$ define the coupling of the leads to the molecules.

The electronic propagator for the coupled system is represented by a retarded Green's function defined as, $G^r(E) = \left[(E + i\eta)I - H_M - \Sigma_L - \Sigma_R\right]^{-1}$, where iη and $\Sigma_{L/R}$ are an infinitesimal imaginary value and the self-energy elements which include the influence of the electrodes, respectively. The conductance of a junction at low bias and coherent regime is then obtained via: $G(E) = \frac{2e^2}{h} Tr\left(G^r \Gamma_L G^a \Gamma_R\right)$ where $\Gamma_{L/R}$ represents the broadening function given by $\Gamma_{L/R}(E) = i(\Sigma_{L/R} - \Sigma_{L/R}^\dagger)$.

The Hamiltonian of the 1D layer containing N molecules is:

$$H_M = \begin{bmatrix} H_M^1 & & \\ & \ddots & \\ & & H_M^N \end{bmatrix} \quad (2)$$

where $H_M^j = \begin{bmatrix} \varepsilon & te^{i\theta} \\ te^{i\theta} & \varepsilon \end{bmatrix}$ is the Hamiltonian of the j-th two-site molecule, with phase-dependent coupling between sites, where θ for simplicity can be either 0 or π.

The Hamiltonians for the 2D leads, $H_{L/M}$, are calculated by assuming 1D columns (parallel to the junction) with an Hamiltonian α coupled to neighboring columns by a coupling matrix β, which in the case of leads with, for example, three lateral sites look like:

$$\alpha = \begin{bmatrix} \varepsilon & t_0 & 0 \\ t_0 & \varepsilon & t_0 \\ 0 & t_0 & \varepsilon \end{bmatrix} \text{ and } \beta = \begin{bmatrix} t_0 & 0 & 0 \\ 0 & t_0 & 0 \\ 0 & 0 & t_0 \end{bmatrix}. \quad (3)$$

The self-energy elements are $\Sigma_{L/R} = V_{L/R}g_{L/R}V_{L/R}^\dagger$, where $g_{L/R}$ are the surface Green function calculated iteratively by:

$$g_{L/R}^{-1} = \left(E + i^{0+}\right)I - \alpha - \beta^\dagger g_{L/R}\beta \quad (4)$$

where $0^+$ is a positive infinitesimal.

The $\varepsilon_n$'s that are the eigenvalues of α, are given analytically by: $\varepsilon_n = \varepsilon - 2t_0\cos(k_n a)$ with $k_n a = \frac{n\pi}{N+1}$. They represent the bottom energies of the sub-band modes within the leads.

In the calculations presented in Fig. 6, the Fermi energy, $E_F$, was chosen to equal certain values within the set of $\varepsilon_n$'s in order to ensure the indicated number of modes within the leads. In all calculations $E_F$ on the Au side was placed at the middle of the band with 100 modes in this lead. The simulations show that the large number of modes in the Au lead does not affect the interference properties of the junctions. This is expected based on the simple reasoning that the contact resistance between two leads with N and M transverse modes is: $R \sim h/2e^2 \left(\frac{1}{M} + \frac{1}{N}\right)$. Since for the given geometry the number of modes in Bi is orders of magnitude smaller than in Au, the entire behavior of the junctions is governed by the Bi lead.

Since each mode has a spatial shape at the molecular interface, the conductance of a junction with a certain combination of $(n_\pi, n_0)$ should also depend on the spatial distribution of the molecules. Therefore, for a given number of $n_\pi$ the conductance was calculated 100 times, each with a random distribution of the position of the molecules. The reported values are the average transmission values of these 100 calculations per each $(n_\pi, n_0)$ combination.

## Data availability
The authors declare that the main data supporting the findings of this study are available within the article.

## Code availability
Code used for the simulations described in this study are available from the corresponding author on reasonable request.

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

## Acknowledgements

Support by the Israel Science Foundation (grant number 1059/18) and the Schmidt Futures foundation (via a breakthrough science grant) to Y. S. are gratefully acknowledged.

## Author contributions

Y.S. conceived and supervised the project. P.L. designed and performed the experiments. Y.S. performed all simulations. P.L. and Y.S. analyzed the data and wrote the manuscript.

## Competing interests

The authors declare no competing interests.
