## [Peer Review File · Nature Communications]

Molecular Ensemble Junctions with Inter-molecular Quantum InterferenceREVIEWER COMMENTS

Reviewer #1 (Remarks to the Author):

This is an interesting study with high novelty for its exploitation of quantum interference in molecular ensemble SAM junctions and the deployment of Bi top contact that give the very long coherence lengths of electrons and thereby facilitate the QI effects in the ensemble. So I do like the concept. However, the study is highly predicated on the emergence of bimodality in the experimental conductance data for the nanopore junctions. The bimodality is sufficiently convincing for the C10 junctions in Figure 3 but the fits of Figure 3a (C6) and Figure 3b (C8) to bimodality might be questioned. If the authors can somehow strengthen this it would greatly enhance the hypothesis. Other minor points are:

- Little is said about the surface chemistry of bismuth. The Bi top contacts are thermally evaporated under vacuum. Do the conditions for fabrication and testing retain Bi in an oxide free state? The DOS might depend sensitively on the nature of the Bi surface. Can that be commented on?
- For ensemble junctions fabricated either by nanopore or other larger area junction methods it is expected that only a proportion of the molecules are "properly contacted". Can a section be given over to discuss this?
- IETS might be presented with bias scale transcribed to cm^{-1} .

Reviewer #2 (Remarks to the Author):

Review

Summary:

In this article the authors have shown that quantum interference (QI) in multi-molecular junctions (nanopore molecular ensemble junctions (MEJs)) with Bismuth as the top contact goes beyond intra-molecular interference and instead probes the inter-molecular interaction. These inter-molecular interactions have previously not been thought of as the main contributor for the QI properties. This shows that it is important not to neglect these interactions, as they influence the QI we observe in molecular junctions. Further, they show that Bismuth could be a new substitute for Au as a top electrode in these type of junctions since these junctions can be produced with low damage effect (when evaporating the bismuth on top) and thereby results in a higher yield of working junctions. This is something that is difficult to obtain when using Au, as Au tends to damage the SAM, creating short circuits and thereby resulting in low yields of working junctions.

Overall, this is an extremely interesting study with very original ideas. The biggest problem is that, in its present form, many details are not explained and thus it is very difficult as a reader to follow the ideas being introduced. In my opinion, the paper would be strengthened considerably if the results and discussion were expanded significantly to allow for a much more thorough and pedagogical presentation of the ideas. The paper also includes a significant number of both experimental and theoretical methods, but these are only described extremely briefly. In the present form, I do not believe enough detail is provided to reproduce the results from any part. My initial impression is that the manuscript is likely to be suitable for publication in Nature Communications following (extensive) revisions, but I will need to see the additional details to be provided on revision before I can be confident in this assessment.

Questions:

1. (Figure 1) – I have read this multiple times to be sure, but are the labels of inter-molecular and intra-molecular interference reversed compared with what is in the text? There is something here that is not matching up for me.
2. (Line 62) – Perhaps this is outside the scope of the study, but a question out of interest: have you thought about how/if the uneven junction (Au-mol-Bi) affect the transmission/interference pattern? ie uneven numbers of carbon atoms. Would you expect an effect
3. (Line 85 / Figure 2c) – I cannot seem to find anywhere in the methods that describes how you measure this temperature dependence? At line 85 you mention the figure but not how this was measured.
4. A general comment – almost all figures have font sizes that are hard to read. Eg in axis labels, it would be an advantage to have these increased and also bolded in some cases
5. (Figure 2c) The C10 results show clear variation in the conductance (although each are stable with temperature) between what, I presume, are repeat measurements, this does not seem to be commented on in the caption or the text. It would be useful to have some comment on the reproducibility in this context.
6. (Line 98 / Figure 3a) – How was this line fitted, and how good is the fit, especially for C6 – are there two peaks here? From the plot alone, the 2nd peak is not at all obvious.
7. (Figure 3a) – It could be helpful here to indicate the GH GL peaks on the graphs together with bigger fonts.
8. (Figure 3c) –Bigger fonts and legends would make this easier to interpret. How are the error bars determined?
9. (Line 142-194) Text discussing figure 4 could use a deeper explanation. Beyond it being clear that something is calculated that shows the phase ratio between the molecules can change the overall conductance, it is extremely difficult to conclude any more details from Figure 4. The method describing the results in 4d appears only to appear in the figure caption (?), more detail would be very useful.
10. (Line 168) – Should this be 14 instead of 13 to match the text? If so, the calculation seems to be wrong. $\sqrt{40000/13} = 55$ and $\sqrt{40000/14} = 53$.
11. (Line 169) – I am not sure I understand why this is so?
12. (Line 175) – Reference 56 state that the change in tunneling phase is due to different lengths of alkanes. Here the focus is on alkanes of the same length having the same feature. I am not sure I understand how these phases should change in a SAM of same length alkanes. Wouldn't they have the same phase?
13. (Lines 173 – 175) – More generally, this discussion of possible explanations for the randomization of the phase seems to be a list of possible hypotheses rather than an explanation. My impression is that the language is somewhat too definite given that this is largely speculation. Eg the following discussion from line 176 onwards seems better suited to a model of structural defects (eg cis defects) but the paragraph starts out (line 173) by saying that the randomization is “most likely rooted in inter-chain through space tunnelling”.
14. At the end of the discussion, I am left with a question as to why this model of phase randomization should lead to two peaks in the conductance? Maybe this is my misconception. I could understand why this model would lead to a skewed histogram (with the skewness changing with length) but what is the argument for a second peak? Is this saying that the system is characterized by some particular level of disorder in the phase? Why would this be?
15. (208) – To replicate the results it would be useful having a better description of the

fabrication of the device, e.g., how the SAM was assembled (Was it assembled by laying in a solution for x days? How are you certain that your SAM is not filled with free/unbound molecules? Was it rinsed in clean solution after assembling?)

Reviewer #3 (Remarks to the Author):

The ms. described elegant and fascinating work on a mesoscopic junction with a prototypical molecule type, alkyl di-thiols of intermediate length (before they form loops), on Au. The special features are the use of the earlier reported nanopore approach (well before ref. 30, already in 1997, as can be found for example, in ref. 17) and, especially, the use of vacuum evaporation (as also earlier reported, BUT NOT for Bi!).

The choice of Bi lies at the very heart of the work and while the possibility was noted earlier, this was not at all for the reason the authors chose it here. Thus, while the referencing to earlier work is a bit sloppy, the originality of the work makes that no more than a minor, slightly annoying feature.

The bimodal conductance result is quite amazing if the interpretation is correct. That is a very weighty reason to publish the work so that others in the field can put their teeth into this idea and check it (for example, might this work with ultra-clean graphene?).

The authors make a good case for their quantum explanation. Because, as far as the reviewer knows, this journal does not have a word limit, it is not clear why the introductory remarks in the manuscript's first paragraph are written in so terse a way that it makes it not really a good gateway for reading the ms. for non-specialists.

One wonders how many earlier results from the many labs that work on MEJs, showed some sign of what the authors find, which then were averaged out to provide nice-looking results for publication.

Thus, also to stimulate others to take a fresh look at their data, the ms. should be published.

Some revisions, most of which appear as remarks in the first part, can help to improve its clarity and readability.

We thank all reviewers for appreciating the importance of the manuscript. Their valuable questions and comments helped us (so we hope) to clarify and improve the discussion on crucial points. We address all of them below point by point. Changes in the manuscript are highlighted in yellow.

Reviewer #1

This is an interesting study with high novelty for its exploitation of quantum interference in molecular ensemble SAM junctions and the deployment of Bi top contact that give the very long coherence lengths of electrons and thereby facilitate the QI effects in the ensemble. So I do like the concept. However, the study is highly predicated on the emergence of bimodality in the experimental conductance data for the nanopore junctions. The bimodality is sufficiently convincing for the C10 junctions in Figure 3 but the fits of Figure 3a (C6) and Figure 3b (C8) to bimodality might be questioned. If the authors can somehow strength this it would greatly enhance the hypothesis.

We elaborate in the paper why for shorter alkane chains it becomes more difficult to resolve the bi-modality. Briefly, we are claiming that for short chains, definitely for liquid-like monolayers formed by molecules such as C6, the phase is completely random and only occasionally there are large enough uniform ensembles that can result in measureable constructive QI. We have re-examined the data and added some more results, which now we believe make the bi-modality clearer.

- Little is said about the surface chemistry of bismuth. The Bi top contacts are thermally evaporated under vacuum. Do the conditions for fabrication and testing retain Bi in an oxide free state? The DOS might depend sensitively on the nature of the Bi surface. Can that be commented on?

This is indeed an important point. We have made sure that the Bi films are oxygen free by analyzing the crystalline structure of Bi films evaporated under the same conditions used for the preparation of the junctions. The films were found to be pure, <111> oriented and with sufficiently large grains that very few of them were sufficient to cover the entire top surface of the contacts. All this now appears in the paper.

- For ensemble junctions fabricated either by nanopore or other larger area junction methods it is expected that only a proportion of the molecules are “properly contacted”. Can a section be given over to discuss this?

We find it highly encouraging that the normalization of the conductance of the junctions by the value of the approximated number of molecules works so well for all molecules. This comes as another proof for the reproducibility of the fabrication scheme, basically suggesting that (almost) all assembled molecules are ‘properly contacted’.

FTIR measurements of the initial steps of direct evaporation of metal atoms onto monolayers (see for example App. Phys. Lett. 2004, 84, 4008) suggest that the first layers of evaporated atoms cover the monolayers quite uniformly. The experimental conditions used in these experiment also prevailed here and are different from the conditions used by others in order to improve the yield: (a) the junctions were not cooled down during the evaporation process in order to protect the molecules form damage and (b) the cross section of the junctions was not made to be smaller than the average grain of Bi.

Maintaining the monolayer at low temperatures while evaporating could indeed improve the yield however with the price that the monolayer is covered by most probably few monolayers of water molecules. As a result, while the yield is improved the number of properly contacted molecules becomes essentially unknown.

Similarly, by using junctions with a small cross sections, leads to accumulation of atoms on the walls of the pores that could block evaporated atoms from reaching the monolayer, resulting again with an unknown number of molecules within the junctions.

Specifically, to Bi, under the conditions used here, Bi forms <111> films with a bilayers structure parallel to the surface. We believe that by not cooling the junctions while evaporating, we allow the initial layers of atoms to properly arrange themselves, and minimize energy, at the interface with the molecules. A similar process is known to take place when ordered Bi₂S₃ layers is formed on top of Bi <111>.

All this discussion has been added into Methods.

- IETS might be presented with bias scale transcribed to cm-1.

A top cm⁻¹ scale has been added to the IETS plot.

Reviewer #2 (Remarks to the Author):

In this article the authors have shown that quantum interference (QI) in multi-molecular junctions (nanopore molecular ensembles junctions (MEJs)) with Bismuth as the top contact goes beyond intra-molecular interference and instead probes the inter-molecular interaction. These inter-molecular interactions have previously not been thought of as the main contributor for the QI properties. This shows that it is important not to neglect these interactions, as they influence the QI we observe in molecular junctions. Further, they show that Bismuth could be a new substitute for Au as a top electrode in these type of junctions since these junctions can be produced with low damage effect (when evaporating the bismuth on top) and thereby results in a higher yield of working junctions. This is something that is difficult to obtain when using Au, as Au tends to damage the SAM, creating short circuits and thereby resulting in low yields of working junctions.

Overall, this is an extremely interesting study with very original ideas. The biggest problem is that, in its present form, many details are not explained and thus it is very difficult as a reader to follow the ideas being introduced. In my opinion, the paper would be strengthened considerably if the results and discussion were expanded significantly to allow for a much more thorough and pedagogical presentation of the ideas. The paper also includes a significant number of both experimental and theoretical methods, but these are only described extremely briefly. In the present form, I do not believe enough detail is provided to reproduce the results from any part. My initial impression is that the manuscript is likely to be suitable for publication in Nature Communications following (extensive) revisions, but I will need to see the additional details to be provided on revision before I can be confident in this assessment.

The paper is now much more detailed both in the experimental sections and in the discussion and models. All added text is marked.

Questions:

1. (Figure 1) – I have read this multiple times to be sure, but are the labels of inter-molecular and intra-molecular interference reversed compared with what is in the text? There is something here that is not matching up for me.

Thank you for noticing this error. The labels are now correct.

2. (Line 62) – Perhaps this is outside the scope of the study, but a question out of interest: have you thought about how/if the uneven junction (Au-mol-Bi) affect the transmission/interference pattern? I.e. uneven numbers of carbon atoms. Would you expect an effect?

The contact resistance between two leads with N and M transverse modes is: $R \sim h/2e^2 \left(\frac{1}{M} + \frac{1}{N} \right)$. Since for the given geometry the number of modes in Bi is orders of magnitude smaller than in Au, the entire behavior of the junctions is governed by the Bi lead. The simulations in the paper now account for the fact that one of the leads is Au. As expected, the resulting behavior is identical to the one presented in the previous version of the manuscript. We comment on this in the manuscript and describe in Methods how the simulations account for the asymmetry. As for the behavior of uneven numbers of carbons, we are currently in the process of measuring such junctions.

3. (Line 85 / Figure 2c) – I cannot seem to find anywhere in the methods that describes how you measure this temperature dependence? At line 85 you mention the figure but not how this was measured.

Details regarding this measurement have been added into Methods. The text now reads: Temperature dependent conductance measurements were performed by a lock-in technique constantly probing the conductance of junctions around a dc bias of 0V, with a modulation ac voltage of 3mV at 137Hz while slowly varying the temperature between 8-300K over a time duration of ~20 hours.

4. A general comment – almost all figures have font sizes that are hard to read. Eg in axis labels, it would be an advantage to have these increased and also bolded in some cases.

All figures have been amended to comply (so we hope) with the guidelines for the preparation of figures.

5. (Figure 2c) The C10 results show clear variation in the conductance (although each are stable with temperature) between what, I presume, are repeat measurements, this does not seem to be commented on in the caption or the text. It would be useful to have some comment on the reproducibility in this context.

Additional details regarding the presented data has been added into the figure caption: The effect of temperature on the conductance of three different C8 (magenta) and three different C10 (blue) junctions in the temperature range between 8-300K. The variation between junctions comply with the variation in conductance as presented in all other figures. The presented behavior for each junction was found to be highly reproducible under repeated measurements and temperature cycles.

6. (Line 98 / Figure 3a) – How was this line fitted, and how good is the fit, especially for C6 – are there two peaks here? From the plot alone, the 2nd peak is not at all obvious.

Multiple Gaussian peaks were fitted (using an OriginLab software) initially to the C10 and C8 data, which show a clear-to-the-eye bi-modal behavior. The high conductance peak of C6 would have been ignored by us if no bi-modal behavior had been observed with C8 and C10. We elaborate in the text why the statistics of the high conductance peak of C6 is so poor. But still, since it fits the expected tunneling attenuation behavior as a function of length, it cannot be ruled out as noise.

7. (Figure 3a) – It could be helpful here to indicate the GH GL peaks on the graphs together with bigger fonts.

We agree. The peaks are now marked.

8. (Figure 3c) – Bigger fonts and legends would make this easier to interpret. How are the error bars determined?

Fonts have been amended. An explanation how the error bars are determined has been added into the figure caption. The text reads: The error bars of the new results are the standard deviations of the Gaussians in Fig. 3b. The error bars for the SMJs results were taken from the published results appearing in the references mentioned in the text.

9. (Line 142-194) Text discussing figure 4 could use a deeper explanation. Beyond it being clear that something is calculated that shows the phase ratio between the molecules can change the overall conductance, it is extremely difficult to conclude any more details from Figure 4. The method describing the results in 4d appears only to appear in the figure caption (?), more detail would be very useful.

The text has been changed profoundly and is now much more detailed. The relevant paragraphs are marked.

10. (Line 168) – Should this be 14 instead of 13 to match the text? If so, the calculation seems to be wrong. $\text{Sqrt}(40000/13) = 55$ and $\text{Sqrt}(40000/14) = 53$.

Since the measured conductance without molecules is $13G_0$, the experimental number of modes is 13. The number of modes approximated by the Sharvin formula, 14, is in very good agreement with the experimental finding, considering the involved geometry and some imperfection induced by the crystalline structure. We have used the experimental value in order to estimate the number of molecules that are coherently probed by each mode. By using the theoretical number of modes, the picture effectively remains the same (a change from 55 to 53 molecules is marginal): in both cases we are still in a regime where inter-molecular QI prevails.

11. (Line 169) – I am not sure I understand why this is so?

This line refers to the main idea of this paper, and we hope can now be better understood after modifying the relevant text (see our answer above in point # 9). Briefly, as long as the number of modes in the leads is smaller than the number of participating molecules in the junction, we expect to be in the inter-molecules QI regime.

12. (Line 175) – Reference 56 state that the change in tunneling phase is due to different lengths of alkanes. Here the focus is on alkanes of the same length having the same feature. I am not sure I understand how these phases should change in a SAM of same length alkanes. Wouldn't they have the same phase?

13. (Lines 173 – 175) – More generally, this discussion of possible explanations for the randomization of the phase seems to be a list of possible hypotheses rather than an explanation. My impression is that the language is somewhat too definite given that this is largely speculation. Eg the following discussion from line 176 onwards seems better suited to a model of structural defects (eg cis defects) but the paragraph starts out (line 173) by saying that the randomization is “most likely rooted in inter-chain through space tunnelling”.

We address the above two points (12 and 13) together:

The reasoning behind choosing alkane-chains for this research is a compromise between two competing demands that are necessary for this research at this stage. Since we wish to present the existence of inter-molecular QI in MEJs, it is important to use molecules for which on one hand their tunneling phase can be varied (even if at this stage not deterministically) and on the other hand the cross-talk between them, once they are assembled in an MEJ, would be minimized in order to ensure that this process does not mask the phase (QI) effect. The cross-talk between saturate alkane-chains is for any practical reasons effectively zero (in the absence of pi-electrons). On the other hand, calculating the tunneling phase through alkanes as a function of their conformation is an elaborate task, which has never been properly done. The literature, however, describes implicitly in several studies (references mentioned in the text) why a phase change is expected in these molecules both as a function of conformation and also due to their assembly in a layer (inter-molecule tunneling). We agree that without any definite theoretical study regarding this phase the discussion should be based on a more suggestive and less definite language. We have modified the text in this way. Please refer to the relevant paragraphs to see all changes.

14. At the end of the discussion, I am left with a question as to why this model of phase randomization should lead to two peaks in the conductance? Maybe this is my misconception. I could understand why this model would lead to a skewed histogram (with the skewness changing with length) but what is the argument for a second peak? Is this saying that the system is characterized by some particular level of disorder in the phase? Why would this be?

See the added explanation in the last two pages of the manuscript. We try to help the readers to follow the implications of inter-molecular QI by suggesting that one should think of the junctions as if they contain a small number of ‘effective molecules’ that due to the fact that their number is small a bi-modal behavior as in junctions containing very few molecules can be observed. By ‘effective molecules’ we refer to the small ensembles of molecules within the junctions that are coherently probed by the long wavelength modes.

15. (208) – To replicate the results it would be useful having a better description of the fabrication of the device, e.g., how the SAM was assembled (Was it assembled by laying in a solution for x days? How are you certain that your SAM is not filled with free/unbound molecules? Was it rinsed in clean solution after assembling?)

All necessary details now appear in Methods.

Reviewer #3 (Remarks to the Author):

The ms. described elegant and fascinating work on a mesoscopic junction with a prototypical molecule type, alkyl di-thiols of intermediate length (before they form loops), on Au. The special features are

the use of the earlier reported nanopore approach (well before ref. 30, already in 1997, as can be found for example, in ref. 17) and, especially, the use of vacuum evaporation (as also earlier reported, BUT NOT for Bi!).

The choice of Bi lies at the very heart of the work and while the possibility was noted earlier, this was not at all for the reason the authors chose it here. Thus, while the referencing to earlier work is a bit sloppy, the originality of the work makes that no more than a minor, slightly annoying feature.

As the experimental part of the manuscript is now more detailed, we have also modified the references relevant to this part. Specifically, as a tribute to Prof. Mark Reed, we added his 1997 paper, which introduced nanopore molecular junctions for the first time.

The bimodal conductance result is quite amazing if the interpretation is correct. That is a very weighty reason to publish the work so that others in the field can put their teeth into this idea and check it (for example, might this work with ultra-clean graphene?).

The authors make a good case for their quantum explanation. Because, as far as the reviewer knows, this journal does not have a word limit, it is not clear why the introductory remarks in the manuscript's first paragraph are written in so terse a way that it makes it not really a good gateway for reading the ms. for non-specialists.

We have expanded the introduction a little bit to give a broader perspective regarding the role of phase in nano-scale transport.

One wonders how many earlier results from the many labs that work on MEJs, showed some sign of what the authors find, which then were averaged out to provide nice-looking results for publication.

Thus, also to stimulate others to take a fresh look at their data, the ms. should be published.

This is a good point. We strongly believe that previous measurements based on monolayers on semiconductors as leads, in which the Fermi wavelength is long, were very likely affected by inter-molecular QI.

We agree that the implications and options for future research are VERY exciting. We are currently pushing this in various routes, focusing at this stage on Bi surfaces. The use of graphene is an obvious excellent possibility, however, experimentally very challenging.

Some revisions, most of which appear as remarks in the first part, can help to improve its clarity and readability.

See all our answers above and all modifications in the manuscript.

Editorial Note: In the review of the second version of this manuscript, reviewer #3 added the comments for modifications to the manuscript file instead of a separated report.

REVIEWER COMMENTS

Reviewer #2 (Remarks to the Author):

I would like to thank the authors for all the effort they have gone to in adding extra explanation and discussion. My impression is that the manuscript is very much improved, and I hope they agree. I have no further comments and recommend publication.

Reviewer #3 (Remarks to the Author):

my comments on the original version were not(well) addressed and there are gaps between what is written in the response and what is done in the revision.

For some reason we did not see the manuscript file with the in-text comments made by the reviewers. Here, we address them all point by point. The corresponding changes/additions in the text are marked in yellow.

We wrote “We are not aware of any other approach to produce MEJs that is based on direct evaporation of a top contact with such a high yield”.

In response, the reviewer suggests several references in which similar and even higher yields are reported.

Admittedly, the definition of yield in this sentence was not clear as our intention was to suggest, following reference 31, that our approach enables high yield based on a process that is prone to mass production, which is unlike many of the methods appearing in the references mentioned by the reviewer. In addition, we also have critical reservations regarding some of the reported yields in these previous studies. Having said all that, since the **comparison of yields** is not the main subject of this paper the sentence was simply erased. We leave it to the readers to decide if the reported yield and the presented fabrication approach are sufficiently impressive to be used in other studies. But still, since the high yield is critical for the large volume of statistics that is necessary for the interpretation (as mentioned in the text) the method section still comments about the advantage of using Bi for the preparation of nanopore junctions by direct evaporation.

We wrote: “Fig. 3a shows the fingerprint region in the spectra of representative C8 and C10 junctions along with the assignment of the vibrational modes, which agrees very well with previous studies³³”.

In response, the reviewer commented that: "What this shows is that there are organic molecules in the junction. The reference shows spectra (Figs, 2 and 3) that do not quite match the one shown here; esp. missing is the 360 meV feature. In any case as presented the method may well damage the top of the molecules but certainly leaves enough intact to generate an IETS results and that is what is important as the point of the ms. need not be that the molecules remain intact, suffice that significant continuous parts of them remain."

Figure 3 now contains the 360meV peak, which appears in all junctions. We believe it is more important to focus on the fingerprint region of the spectra, which is more informative as has been done by others (references 31 and 33). The matching of the peaks in this region to the two previous studies is good (with a variation in some cases of few meV) considering the fact that it is well known that the contacting metal affects the frequency of the vibrational modes. What is unique in our system is the (high) intensity of some of the modes. We do not discuss this point further here since it does not contribute to the main theme of the paper and because further work needs to be done to understand this behavior (work is currently in progress). We believe that the unusual high intensity of some modes results from the low number of carriers and states within the Bi, which reduce the probability for annihilation of excited vibrational modes via creation of electron-hole pairs within the metal.

In response to the sentence: "The histograms show a double-peak distribution (a bi-modal behavior), with high, G^H , and low, G^L , conductance average values, which become better resolved as the length of the molecules increases, i.e., their statistics is a molecular property."

The reviewer asks about the statistics as a molecular property: "Why? It can be a spatial feature, i.e., distance between electrodes."

In the used nanopore junctions, the length of the molecules defines the distance between electrodes, and therefore it results from a molecular property. As we discuss further below in the manuscript, it is not just the molecular property of length that is important here, but instead the order and uniformity within the monolayers, which results from this length that affects the overall behavior.

The reviewer comments that the ratio DOS_{Bi}/DOS_{Au} is $\sim 24,000$ and not $\sim 23,000$.

Correct, the number is now 24,000. In the calculations, we used the exact value of 23,810.

The reviewer questions where "...the difference between their corresponding G_{SMJ}^H and G_{SMJ}^L values" is observed.

The references of all previous SMJ measurements of alkane chains were cited again, in which G_{SMJ}^H and G_{SMJ}^L are reported.

The reviewer suggests that the paragraph starting with "Intermolecular QI is demonstrated in ..." should be changed to be less definite.

We comply and the text now reads: "The results in Fig. 5 and Fig. 6 are consistent with *Inter-molecular* QI."

The reviewer wrote: "not clear why abbreviations that are used only ONCE in the abstract need to be used there. Keep Abstract simple and then define them in the text".

We accept the suggestion, the abbreviations were omitted from the abstract and only appear in the main text.

We wrote: "An electron wave transmitted through a conductor's orbital acquires a phase, θ , which provides information complementary to the transmission probability $T = |t|^2$, with t being the transmission amplitude^{1,2}: $t = \sqrt{T}e^{i\theta}$."

The reviewer requested to add why this is interesting/important.

The following sentence was added: "Control and manipulation of this phase enables to modify transport properties by quantum interference (QI)."

The rest of the introductory paragraph explains this point in detail.

We wrote: "The use of alkane chains insures that no crosstalk exists between the assembled molecules...."

The reviewer believes that a less definite sentence would be more appropriate.

Therefore, the text now reads:

"It is very unlikely that crosstalk exists between such molecules once assembled, which is important since such a process could potentially alter or totally mask *inter-molecular* QI²¹."

The reviewer is concerned about the possible effect of the thick amorphous Bi oxide layers on the electronic properties of the films.

Since Bi is directly evaporated on the monolayer within the pores, an oxide layer exists only at the top surface of the films. We are not aware of any study that reports of a change in the Fermi wavelength of relatively thick films, such as the ones used here (60nm), due to oxide layers. On the contrary, recent calculations of free standing very thin films (3 bilayers ~ 1 nm) show that the oxide layers do not affect

the electronic structure at all. Text regarding this point has been added in Methods including the reference describing the calculation.

In the discussion we report that the conductance of direct Bi-Au contacts is $13G_0$. The reviewer asks which metal fill the pores.

As explained in the fabrication section of Methods, Bi is evaporated to fill the pores and therefore no oxide is formed at the (buried) interface between Bi and either the underlying Au (in the case of direct contact) or the molecules (in the case of MEJs).

The reviewer asks regarding figure 6a, how the experimental results refer to the theoretical curves in this figure.

The text has been changed to hopefully make the relation clearer. The text now reads:

"Since, one may regard this ratio as the effective number of molecules that is coherently and simultaneously probed by the tunneling electrons in these junctions, the experimental ratio of ~ 55 corresponds to the 2-modes (second from the bottom) curve in Fig. 6a, which was calculated for 100 molecules."

Regarding Fig 7, the reviewer correctly note that n should be replaced by $N_{molecules}$ in the figure caption and also asks for a better explanation why the conductance according eq. 1 is proportional to $N_{molecules}^2$ in the case of uniform phase.

The text now reads:

Returning to Eq. 1., the red curve in Fig. 7, shows the calculated conductance based on this equation of an MEJ with $N_{molecules} = 1000$ and one transverse mode at the Fermi level as a function of the $n_\pi:n_0$ ratio ($N_{molecules} = n_\pi + n_0$). In this calculation, since there is only one transverse mode, $N_{molecules}$ defines the number of interfering transmission paths. It is then easy to see by performing the summations in Eq. 1 up to $N_{molecules}$, that when the phase is identical for all paths (a situation which occurs on both ends of the red curve) the conductance is proportional to $N_{molecules}^2$.

Towards the end of the manuscript the reviewer asks regarding the use of reference 39 in relation with the explanation about "effective molecules".

Indeed, the text should have been written differently. It now reads:

"MEJs with leads having a long Fermi wavelength, although containing thousands of molecules, essentially behave like junctions with very few molecules, in which each 'effective molecule' is a small ensemble probed by individual electrons having a long characteristic wavelength. One can consider this as a loss of resolution: with increasing wavelength of the tunneling electrons, the individual transmission properties of the molecules cannot be resolved anymore and the overall conductance starts to depend on the size of the ensembles that are being tunneled-through which in turn depends on the number of modes. When the number of "effective molecules" is sufficiently small the junction starts to behave similarly to a small-ensemble junction³⁹ with a bi-modal conductance behavior."

The reviewer suggests to modify caption of figure 2f.

The text now reads:

The diffraction pattern of the red rectangle in (e), which is typical for all measured grains, reveals Bi metal with a $\langle 111 \rangle$ orientation.

REVIEWERS' COMMENTS

Reviewer #3 (Remarks to the Author):

No further comments and the work should be published. The readers will hopefully figure it out.